# Climate change exacerbates hurricane flood hazards along US Atlantic and Gulf Coasts in spatially varying patterns

Reza Marsooli [1], Ning Lin [2], Kerry Emanuel [3] & Kairui Feng [2]

One of the most destructive natural hazards, tropical cyclone (TC)–induced coastal flooding, will worsen under climate change. Here we conduct climatology–hydrodynamic modeling to quantify the effects of sea level rise (SLR) and TC climatology change (under RCP 8.5) on late 21st century flood hazards at the county level along the US Atlantic and Gulf Coasts. We find that, under the compound effects of SLR and TC climatology change, the historical 100-year flood level would occur annually in New England and mid-Atlantic regions and every 1–30 years in southeast Atlantic and Gulf of Mexico regions in the late 21st century. The relative effect of TC climatology change increases continuously from New England, mid-Atlantic, southeast Atlantic, to the Gulf of Mexico, and the effect of TC climatology change is likely to be larger than the effect of SLR for over 40% of coastal counties in the Gulf of Mexico.

[1] Department of Civil, Environmental and Ocean Engineering, Stevens Institute of Technology, Hoboken, NJ 07030, USA. [2] Department of Civil and Environmental Engineering, Princeton University, Princeton, NJ 08544, USA. [3] Department of Earth, Atmospheric and Planetary Sciences, Massachusetts Institute of Technology, Cambridge, MA 02139, USA. Correspondence and requests for materials should be addressed to N.L. (email: nlin@princeton.edu)

Coastal flooding responds to both sea level rise (SLR) and storm climatology change. SLR varies from place to place[1–3] due to the ocean circulation and glacial isostatic adjustment, and climate change results in an interbasin variation of cyclone characteristics[4–10]. Recent research has shown that the spatial variability in SLR and extratropical cyclone (ETC) climatology change results in flood hazards that vary across the basin and global scales[11–14]. However, the most destructive coastal floods are caused by tropical cyclones (TCs)[15]. Yet effects of TC climatology change on flood hazards at the basin and global scales have not been investigated, nor have the compound effect of SLR and TC climatology change and its spatial variation. Understanding the basin to global scale variation of TC flood hazards and their future evolution is critical, if we are to identify the current and future degree of flood risk in different regions and to prioritize adaptation and mitigation investments.

Quantifying large-scale TC flood variations in the future climate is challenging for two reasons. First, most reanalysis datasets and global circulation models (GCMs) cannot resolve TCs well due to their relatively low resolutions[16]. Thus, recent studies which assessed global scale flood hazards using these models[11,13] accounted for ETCs well but not TCs. In recent years, a few high–resolution GCMs and regional downscaled climate models have been developed to study TCs in a warming climate[8,9,17]. These models are still computationally expensive and thus not practical for flood hazard assessment studies, which should consider large spectra of storm scenarios in order to address hazards induced by low-probability, high-consequence events[18]. An effective approach is to statistically generate large samples of synthetic TCs for reanalysis or GCM-projected climate conditions[19,20] to drive hydrodynamic modeling and assess flood hazards. This climatological–hydrodynamic approach[21], however, induces the second challenge: balancing accuracy and efficiency in hydrodynamic modeling. In order to accurately estimate floods, hydrodynamic modeling is often performed on high-resolution numerical meshes that can capture complex coastal bathymetry and topography. Thus, the climatological–hydrodynamic approach, which requires a large number of simulations on the computationally expensive numerical meshes, has mainly been applied at city or regional scales[18,22].

Here we investigate the effects of SLR and TC climatology change on future flood hazards along the entire US Atlantic and Gulf Coasts. To do so, we apply the climatological–hydrodynamic approach[21] at the basin scale. Specifically, we use a statistical/deterministic hurricane model[19] to generate large numbers of synthetic TCs under historical (1980–2005) and future projected (2070–2095) climate conditions for the Atlantic basin. We apply a widely used hydrodynamic model[23,24] with a recently developed basin scale computational mesh[25] to simulate the storm tides (the combination of storm surge and astronomical tide) induced by these synthetic TCs for the US Atlantic and Gulf Coasts. Then we estimate the historical and future return periods of flood heights, defined as the combination of storm tide and SLR (based on a probabilistic projection[2]), for each county along the US Atlantic and Gulf Coasts. In particular, we examine the spatial variation of the flood return levels and relative impacts of SLR and TC climatology change along the US Atlantic and Gulf Coasts. We find that, under the compound effects of SLR and TC climatology change, the historical 100-year flood level would occur annually in New England and mid-Atlantic regions and every 1–30 years in southeast Atlantic and Gulf of Mexico regions in the late 21st century. The relative effect of TC climatology change increases continuously from New England, mid-Atlantic, southeast Atlantic, to the Gulf of Mexico, and the effect of TC climatology change is likely to be larger than the effect of SLR for over 40% of coastal counties in the Gulf of Mexico.

## Results

**Modeling and analysis.** The statistical/deterministic hurricane model[19] generates synthetic TCs for a given large-scale atmospheric and oceanic environment estimated from observations or a climate model (see "Methods" section). We run the model to generate 5018 synthetic TCs for the observed climate of the historical period between 1980 and 2005, based on the National Centers for Environmental Prediction (NCEP) reanalysis[26]. To study the TC climatology change, we run the model to generate synthetic TCs for the projected climate of the future period between 2070 and 2095, under the RCP 8.5 greenhouse gas concentration scenario. Given data availability and following previous studies[7,18], we consider projections from six CMIP5 GCMs[27] including CCSM4 (Community Climate System Model, the University Corporation for Atmospheric Research); GFDL5 (Geophysical Fluid Dynamics Laboratory Climate Model, USA); HadGEM5 (Hadley Centre Global Environment Model, U.K. Meteorological Office); MIROC (Model for Interdisciplinary Research on Climate, University of Tokyo, National Institute for Environmental Studies, Japan, and Japan Agency for Marine-Earth Science and Technology Frontier Research Center for Global Change); MPI5 (Max Planck Institute for Meteorology, Germany); and MRI5 (Meteorological Research Institute, Japan). We simulate 5018 synthetic TCs for each model for the future period. We also generate another 5018 TCs for each model for the historical period.

The storm tide induced by each generated synthetic storm is simulated using the advanced circulation model (ADCIRC)[23,24] with a basin scale mesh[25] (see "Methods" section). For each coastal county, we extract the storm tide associated with each synthetic TC as the largest peak storm tide generated by the TC along the county's coastlines. The probabilistic SLR projection is obtained from ref. [2] (see "Methods" section). For each coastal county, we use the projection under RCP 8.5 of the end-of-21st century SLR from the closest station to that county. Combining the probabilistic projections of storm tide and SLR, we perform statistical analysis to estimate the return periods of flood heights (see "Methods" section). Considering that climate model projections may be biased, we bias correct the climate model-projected storm tides based on a comparison of the model estimates for the historical period with the NCEP-based estimates (see "Methods" section). We also obtain a weighted average projection of storm tides with the weight on each climate model depending on its accuracy in the historical estimations relative to the NCEP estimations (see "Methods" section). The flood return level estimations are performed for each coastal county along the US Atlantic and Gulf Coasts (Fig. 1 shows the list of the counties; the basin is divided into four regions: Gulf of Mexico, southeast Atlantic, mid-Atlantic, and New England).

The hydrodynamic model was previously evaluated against historical storm tides and the model showed a satisfactory performance[25]. The hurricane model was also previously evaluated and was shown to generate synthetic storms that statistically agree with observations[28] and compare well with storms generated by other methods[5,29]. Here we evaluate the integrated climatology–hydrodynamic modeling system by comparing the flood return period estimates derived from the NCEP-based synthetic TCs for the historical period of 1980–2005 to those based on the observed water levels at tide gauge stations for the same period. Comparisons show a good agreement between the modeled and observation-based flood level estimates for relatively short return periods that can be resolved based on the observations during the relatively short historical period, although larger return levels cannot be well resolved from the observations and wide uncertainty bounds exist (Supplementary Fig. 1). To assist the spatial comparison of flood hazards, the flood heights are determined relative to the local mean higher high water.

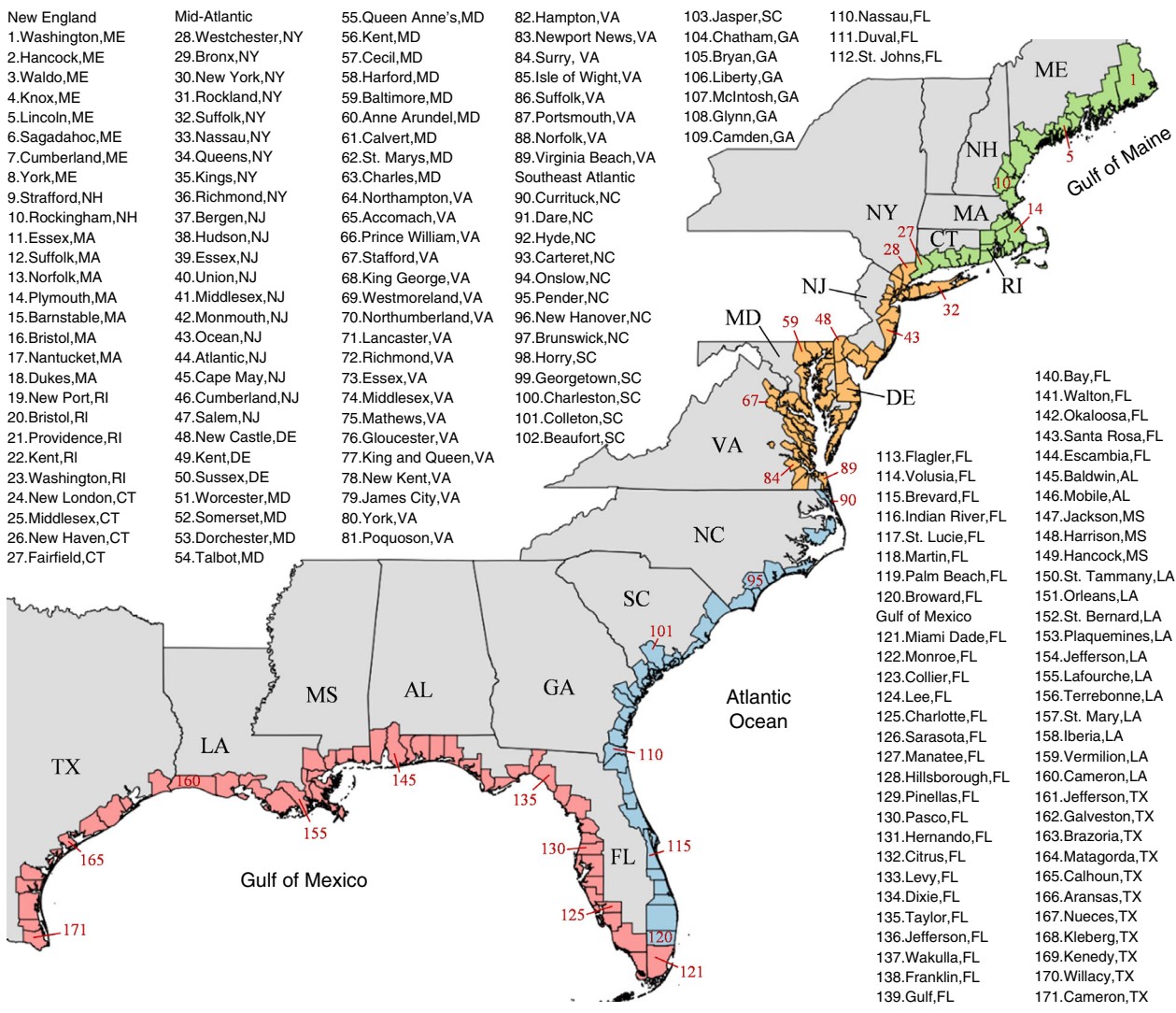

| New England | Mid-Atlantic | 55.Queen Anne's,MD | 82.Hampton,VA | 103.Jasper,SC | 110.Nassau,FL |
|---|---|---|---|---|---|
| 1.Washington,ME | 28.Westchester,NY | 56.Kent,MD | 83.Newport News,VA | 104.Chatham,GA | 111.Duval,FL |
| 2.Hancock,ME | 29.Bronx,NY | 57.Cecil,MD | 84.Surry, VA | 105.Bryan,GA | 112.St. Johns,FL |
| 3.Waldo,ME | 30.New York,NY | 58.Harford,MD | 85.Isle of Wight,VA | 106.Liberty,GA | |
| 4.Knox,ME | 31.Rockland,NY | 59.Baltimore,MD | 86.Suffolk,VA | 107.McIntosh,GA | |
| 5.Lincoln,ME | 32.Suffolk,NY | 60.Anne Arundel,MD | 87.Portsmouth,VA | 108.Glynn,GA | |
| 6.Sagadahoc,ME | 33.Nassau,NY | 61.Calvert,MD | 88.Norfolk,VA | 109.Camden,GA | |
| 7.Cumberland,ME | 34.Queens,NY | 62.St. Marys,MD | 89.Virginia Beach,VA | | |
| 8.York,ME | 35.Kings,NY | 63.Charles,MD | Southeast Atlantic | | |
| 9.Strafford,NH | 36.Richmond,NY | 64.Northampton,VA | 90.Currituck,NC | | |
| 10.Rockingham,NH | 37.Bergen,NJ | 65.Accomach,VA | 91.Dare,NC | | |
| 11.Essex,MA | 38.Hudson,NJ | 66.Prince William,VA | 92.Hyde,NC | | |
| 12.Suffolk,MA | 39.Essex,NJ | 67.Stafford,VA | 93.Carteret,NC | | |
| 13.Norfolk,MA | 40.Union,NJ | 68.King George,VA | 94.Onslow,NC | | |
| 14.Plymouth,MA | 41.Middlesex,NJ | 69.Westmoreland,VA | 95.Pender,NC | | |
| 15.Barnstable,MA | 42.Monmouth,NJ | 70.Northumberland,VA | 96.New Hanover,NC | | |
| 16.Bristol,MA | 43.Ocean,NJ | 71.Lancaster,VA | 97.Brunswick,NC | | |
| 17.Nantucket,MA | 44.Atlantic,NJ | 72.Richmond,VA | 98.Horry,SC | | |
| 18.Dukes,MA | 45.Cape May,NJ | 73.Essex,VA | 99.Georgetown,SC | 140.Bay,FL | |
| 19.New Port,RI | 46.Cumberland,NJ | 74.Middlesex,VA | 100.Charleston,SC | 141.Walton,FL | |
| 20.Bristol,RI | 47.Salem,NJ | 75.Mathews,VA | 101.Colleton,SC | 142.Okaloosa,FL | |
| 21.Providence,RI | 48.New Castle,DE | 76.Gloucester,VA | 102.Beaufort,SC | 143.Santa Rosa,FL | |
| 22.Kent,RI | 49.Kent,DE | 77.King and Queen,VA | | 144.Escambia,FL | |
| 23.Washington,RI | 50.Sussex,DE | 78.New Kent,VA | | 145.Baldwin,AL | |
| 24.New London,CT | 51.Worcester,MD | 79.James City,VA | | 146.Mobile,AL | |
| 25.Middlesex,CT | 52.Somerset,MD | 80.York,VA | | 147.Jackson,MS | |
| 26.New Haven,CT | 53.Dorchester,MD | 81.Poquoson,VA | | 148.Harrison,MS | |
| 27.Fairfield,CT | 54.Talbot,MD | | | 149.Hancock,MS | |

| | |
|---|---|
| 113.Flagler,FL | 150.St. Tammany,LA |
| 114.Volusia,FL | 151.Orleans,LA |
| 115.Brevard,FL | 152.St. Bernard,LA |
| 116.Indian River,FL | 153.Plaquemines,LA |
| 117.St. Lucie,FL | 154.Jefferson,LA |
| 118.Martin,FL | 155.Lafourche,LA |
| 119.Palm Beach,FL | 156.Terrebonne,LA |
| 120.Broward,FL | 157.St. Mary,LA |
| Gulf of Mexico | 158.Iberia,LA |
| 121.Miami Dade,FL | 159.Vermilion,LA |
| 122.Monroe,FL | 160.Cameron,LA |
| 123.Collier,FL | 161.Jefferson,TX |
| 124.Lee,FL | 162.Galveston,TX |
| 125.Charlotte,FL | 163.Brazoria,TX |
| 126.Sarasota,FL | 164.Matagorda,TX |
| 127.Manatee,FL | 165.Calhoun,TX |
| 128.Hillsborough,FL | 166.Aransas,TX |
| 129.Pinellas,FL | 167.Nueces,TX |
| 130.Pasco,FL | 168.Kleberg,TX |
| 131.Hernando,FL | 169.Kenedy,TX |
| 132.Citrus,FL | 170.Willacy,TX |
| 133.Levy,FL | 171.Cameron,TX |
| 134.Dixie,FL | |
| 135.Taylor,FL | |
| 136.Jefferson,FL | |
| 137.Wakulla,FL | |
| 138.Franklin,FL | |
| 139.Gulf,FL | |

**Fig. 1** Coastal counties along the US Atlantic and Gulf Coasts (numbers represent the county ID). The study area is divided into four regions: New England (green), mid-Atlantic (orange), southeast Atlantic (blue), and Gulf of Mexico (red). Source data are provided as a Source data file

**Spatial and temporal variation of flood hazards**. As examples, Fig. 2 shows the return period curves for representative coastal counties in each region. The future return period curves take into account the impacts of SLR and TC climatology change, which is the weighted average over the six climate models (with weighting factors shown in Supplementary Fig. 2). Results indicate that the flood level for a given return period substantially increases by the end of 21st century, due to SLR as well as TC climatology change. The very likely estimates (5th–95th percentiles; i.e., 90% statistical confidence interval) of flood levels with a long return period cover a wide range, indicating a large statistical uncertainty in such events. The uncertainties are smaller for flood levels with a higher probability of occurrence, e.g., the 100-year flood return level. We retain the focus of the remainder of this paper on the 100-year flood level.

Figure 3 displays the spatial distribution of the estimated 100-year flood level along the US Atlantic and Gulf Coasts. The NCEP-based best estimate of 100-year flood level ($\eta_{100-yr}$) for the historical period varies greatly along the coast (Fig. 3a): it is between 1.53 and 4.30 m (with the average over all counties of 3.03 m) in the Gulf of Mexico, 0.52 and 2.82 m (1.46 m) in the southeast Atlantic, 0.27 and 1.67 m (0.84 m) in the mid-Atlantic, and 0.48 and 1.20 m (0.66 m) in the New England regions. Figure 3b shows the spatial distribution of the total changes in

$\eta_{100\,year}$ for the future period, hereinafter $\Delta\eta_{100\,year}$ (changes are weighted average over six climate models; $\Delta\eta_{100\,year}$ projected by each climate model is shown in Supplementary Fig. 3).

Along coastal counties in the Gulf of Mexico region, the best estimate of $\Delta\eta_{100\,year}$ is between 1.5 and 2.80 m, with an average value of 2 m (66% increase in the average $\eta_{100\,year}$). The largest $\Delta\eta_{100\,year}$ is projected to be between 2 and 2.8 m along the northern coast of the Gulf of Mexico (Alabama, Mississippi, Louisiana, and East Texas). The average very likely (5th–95th percentile) range of $\eta_{100\,year}$ changes from 2.38–4.16 m in the historical period to 4.19–6.62 m in the future period for the Gulf of Mexico region. The best estimate of $\Delta\eta_{100\,year}$ is between 1.34 and 1.85 m (average change of 1.52 m; 104% increase) in the southeast Atlantic region and between 1.43 and 1.92 m (average change of 1.67 m; 200% increase) in the mid-Atlantic region. The average very likely estimate of $\eta_{100\,year}$ changes from 1.03–2.32 m to 2.6–3.87 m in the southeast Atlantic region and from 0.65–1.21 m to 2.24–3.10 m in the mid-Atlantic region. The best estimate of $\Delta\eta_{100\,year}$ for the New England region is between 1.55 and 1.82 m (average change of 1.68 m; 255%). The average very likely estimate of $\eta_{100\,year}$ in this region changes from 0.56–0.89 m to 2.23–2.62 m.

A previous study has shown that the TC climatology change and a 1-m SLR by the end of 21st century substantially increase

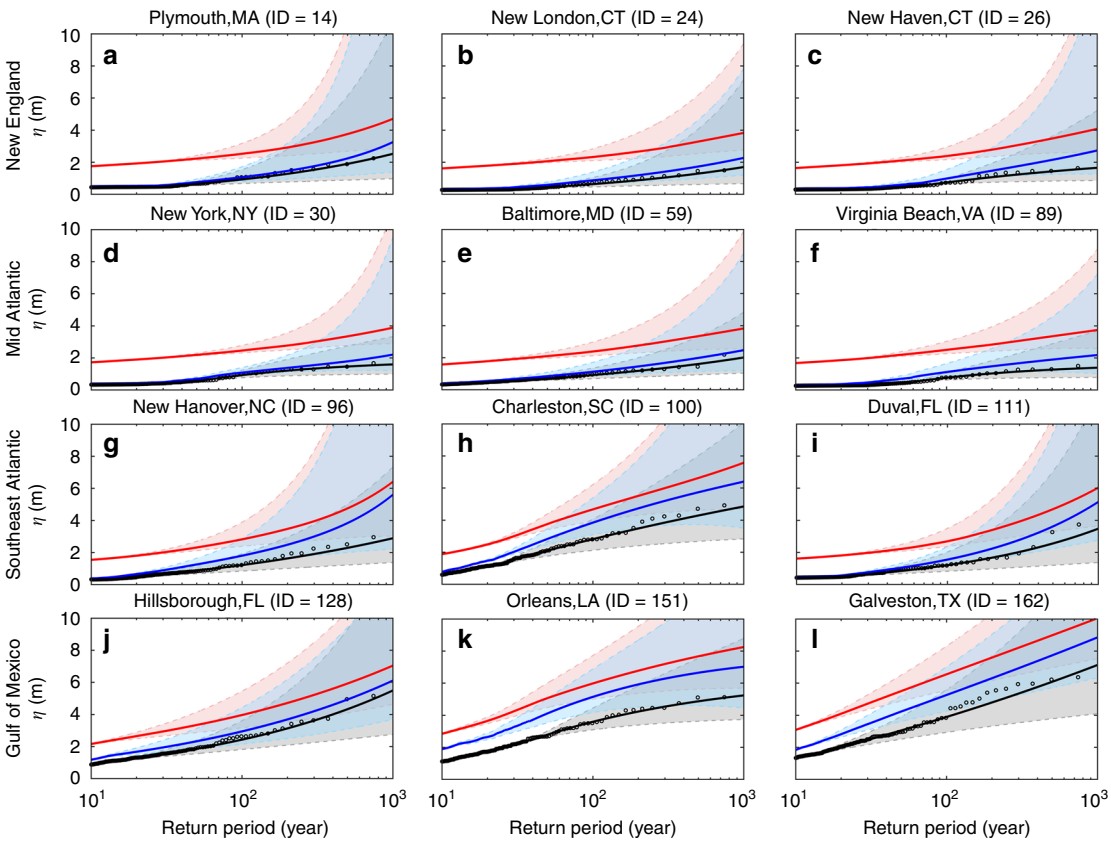

**Fig. 2** Flood return period curves for the historical period of 1980–2005 (black) and future period of 2070–2095 (blue: only effects of TC changes, red: compound effects of SLR and TCs) at selected coastal counties. **a–c** Representative counties in New England; **d–f** representative counties in mid Atlantic; **g–i** representative counties in southeast Atlantic; **j–l** representative counties in Gulf of Mexico. Empirical data points for the historical period are shown as black circles. Solid lines represent the best estimates of flood return periods. Shaded areas cover the very likely range estimates (i.e., 90% statistical confidence interval). Future projections are weighted average over the six climate models. Flood levels are relative to mean higher high water (MHHW, obtained from https://vdatum.noaa.gov). Source data are provided as a Source data file

flood levels at New York City[21]. Four different climate models (based on CMIP3 A1B scenario) in this previous study[21] projected an increase of between 0.8 and 1.75 m in $\eta_{100 \text{ year}}$. Our projections from six climate models (based on CMIP5 RCP 8.5 scenario) for the New York county show an increase of between 1.36 and 1.90 m with a weighted average increase of 1.53 m. The subtle difference between the range of projections is mainly because the previous study was based on a different synthetic TC dataset, its projections were not bias corrected, and a different computational mesh was used in the hydrodynamic model. The previous study was also based on a deterministic SLR of 1 m, whereas the present study is based on a probabilistic projection of SLR.

Figure 3c shows the future return periods of the historical NCEP-based 100-year flood level, which are estimated to be between 5 and 30 years (16.4 years, averaged over all counties) for the coastal counties in the Gulf of Mexico and between 1 and 29 years (average 8.3 years) in the southeast Atlantic. In the New England and mid-Atlantic regions, the historical $\eta_{100 \text{ year}}$ is estimated to occur annually by the end of 21st century. In these high latitude regions, the historical 100-year flood levels are relatively small and thus significant changes in the future climate lead to substantial reductions of the return periods of such flood levels.

**Relative impact of SLR and TC climatology change**. Figure 4 shows the contribution of SLR and TC climatology change (respectively, $\Delta\eta_{100 \text{ year, SLR}}$ and $\Delta\eta_{100 \text{ year, TC}}$) to $\Delta\eta_{100 \text{ year}}$. The

effect of SLR is largest in the mid-Atlantic and New England regions, and the northern coast of the Gulf of Mexico, consistent with the projected SLR patterns[2]. We find that SLR results in an increase in $\eta_{100 \text{ year}}$ of 1.07 m (35% increase in the average $\eta_{100 \text{ year}}$) in the Gulf of Mexico (averaged over all counties in this region), 1.08 m (74%) in the southwest Atlantic, 1.38 m (165%) in the mid-Atlantic, and 1.58 m (239%) in New England. The effect of TC climatology change varies along the coastlines in a contrary way. The TC climatology change alone increases $\eta_{100 \text{ year}}$ by about 0.93 m (31% increase in the average $\eta_{100 \text{ year}}$) in the Gulf of Mexico, 0.44 m (30%) in the southeast Atlantic, 0.29 m (35%) in the mid-Atlantic, and only 0.1 m (15%) in New England. The largest-projected TC induced change in the 100-year flood level is about 1.5 m for several coastal counties in Mississippi and Louisiana. Our projections show a $\Delta\eta_{100 \text{ year, TC}}$ of about 0.16 m for the New York county, NY, which is consistent with a previous study[21], where results from four climate models showed that the influence of TC climatology change on New York City's $\eta_{100 \text{ year}}$ is between −0.2 m and 0.75 m.

Our projections show that TC climatology change has a minimal impact on $\Delta\eta_{100 \text{ year}}$ at high latitudes whereas its impact on $\Delta\eta_{100 \text{ year}}$ at lower latitudes is as significant as SLR. In the New England region, SLR is projected to contribute between 82.2 and 99.7% to $\Delta\eta_{100 \text{ year}}$, whereas TC climatology change contributes only between 0.3 and 17.8% (for counties bordering the Gulf of Maine, $\Delta\eta_{100 \text{ year, TC}} < 3\%$). The contribution of SLR to $\Delta\eta_{100 \text{ year}}$ is between 67.1 and 96.1% in mid-Atlantic and between 44.6 and 89.8% in southeast Atlantic. It is reduced to 35–80.1% for the Gulf

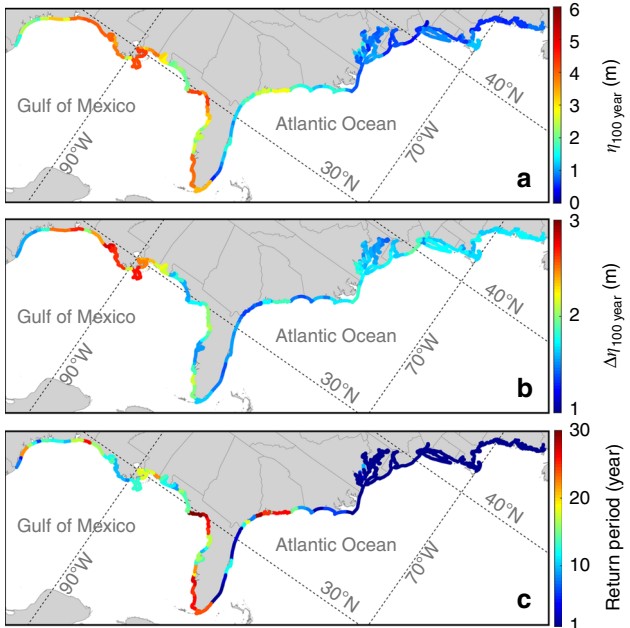

**Fig. 3** Projected flood hazards along the US Atlantic and Gulf Coasts. **a** NCEP-based best estimate of 100-year flood level $\eta_{100\ year}$ for the historical period of 1980–2005. **b** Projected weighted average changes in $\eta_{100\ year}$ for the future period of 2070–2095 under the compound effects of SLR and TC climatology change. **c** Future return period of historical 100-year flood level. Flood levels are relative to MHHW. Source data are provided as a Source data file

of Mexico region. In 41% of coastal counties in the Gulf of Mexico, the TC climatology change is projected to be the main cause of increase in the future 100-year flood level (i.e., contribution of TC climatology change >50%). This spatial trend of relative effects of SLR and TC climatology change on the flood level also exists for other return periods, as shown in Fig. 2 for representative counties.

Increasing flood levels induced by TC climatology change, especially in lower-latitude regions, suggest that the frequency, intensity, and/or size of TCs could increase by the end of 21st century. Figure 5 shows that the frequency, intensity, and size of NCEP-based historical TCs off the US Atlantic and Gulf Coasts greatly varies as a function of latitude. While the TC frequency and intensity (represented by maximum wind speed $V_{max}$) are higher in the Gulf of Mexico and southeast Atlantic regions, the TC size (represented by radius of maximum wind speed $R_{max}$) is larger in the mid-Atlantic and New England regions. Figure 5 shows that both the intensity and size of TCs off the entire US Atlantic and Gulf Coasts increase from the historical period to the future period (up to a 21% increase). In particular, the index $V_{max}^2 R_{max}$, which we use here as a consolidated measure of TC intensity and size, increases in the entire basin with the largest increase in the Gulf of Mexico, resulting in the large values of $\Delta\eta_{100\ year,\ TC}$ projected for the Gulf of Mexico compared with other regions (see Fig. 4). Changes in the TC frequency, shown in Fig. 5, reveal a larger increase for the northern coast of the Gulf of Mexico than the Gulf's eastern and western coasts, explaining the larger $\Delta\eta_{100\ year,\ TC}$ along the coastal counties of Alabama, Mississippi, Louisiana, and East Texas than other counties in the Gulf. In addition, our analysis of TC translation speed, not shown here, reveals an increase in the number of slow-moving TCs and a decrease in the number of fast-moving TCs. Slower TCs allow winds to blow onshore for longer periods of time, resulting in possibly larger and longer coastal flooding.

Projections based on all six climate models agree that the largest impact of TC climatology change on the 100-year flood level takes place in the Gulf of Mexico (Supplementary Fig. 4). Projections from five models (out of six) suggest that $\Delta\eta_{100\ year,\ TC}$ along the northern coast of Gulf of Mexico is larger than that along the Gulf's eastern and western coasts. Only MRI5 projects a larger $\Delta\eta_{100\ year,\ TC}$ along the eastern Gulf Coast. The GFDL5 model shows profoundly larger $\Delta\eta_{100\ year,\ TC}$ in the Gulf of Mexico region. This model projects an average increase of 2.19 m (73% of the average $\eta_{100\ year}$) in this region whereas projections from other models are between 0.49 (16%) and 1.0 m (33%). In the Southeast Atlantic region, HadGEM5 projects the largest $\Delta\eta_{100\ year,\ TC}$ with an average increase of 1.2 m (86%). In this region, MRI5 shows that the impact of TC climatology change on $\eta_{100\ year}$ is between −0.36 (23% decrease) and 0.68 m (57% increase), with an average increase of 0.08 m (8%). The projections from other models are between 0.5 (11%) and 0.79 m (60%). In the mid-Atlantic region, HadGEM5 shows the largest $\Delta\eta_{100\ year,\ TC}$ with an average increase of 0.75 m (90%). For this region, MRI5 shows an average decrease of 0.05 m (4%). Projections from other models indicate an average increase between 0.14 (17%) and 0.53 m (60%). In the New England region, HadGEM5 shows the largest $\Delta\eta_{100\ year,\ TC}$ with an average increase of 0.41 m (57%) whereas MRI5 shows an average decrease of 0.16 m (21%). Other models project an average increase between 0.06 (8%) and 0.13 m (18%).

Discrepancies between $\Delta\eta_{100\ year,\ TC}$ projected by individual climate models (Supplementary Fig. 4) are owing to differences in projections of future TC climatology (see Supplementary Figs. 5–8). For example, projections from GFDL5 show a profoundly larger $\Delta\eta_{100\ year,\ TC}$ in the Gulf of Mexico region, which is due to the substantial increase in all of frequency, intensity, and size of the future TCs projected by this model. However, the skill score and weighting factor of GFDL5 is smaller than other models (Supplementary Fig. 2), leading to a smaller contribution of this model to the weighted average projections discussed earlier.

## Discussion

TC flood risks are evolving along the US Atlantic and Gulf Coasts, owing to SLR and TC climatology change in the western North Atlantic Ocean basin. The rate at which sea level is changing varies from place to place[1–3], affecting future flood hazards locally (e.g., ref. [14]). The impact of TC climatology change in the basin could spatially vary too although, to our best knowledge, this was not investigated prior to the present study. Here, for the first time, we showed that TC climatology change would substantially increase flood return levels, with the highest and lowest impacts in the Gulf of Mexico and Gulf of Maine regions, respectively. We found that the effect of late 21st century TC climatology change on 100-year flood levels exceeds the effect of SLR for over 40% counties along the Gulf of Mexico coast.

A previous study by the United States Army Corps of Engineers, USACE[30] (hereafter NM14), projected the effects of SLR on future flood return levels. Our findings on the effects of SLR on flood return levels are comparable with those from the USACE study. We project, for example, that SLR causes an increase of 1.38 m in the 100-year flood level at the New York county from the historical period of 1980–2005 to the future period of 2070–2095 under the RCP 8.5 scenario. Projections from NM14 for the New York City (The Battery NY) indicate that SLR causes an increase of 0.8 m in the 100-year flood level from 2014 to 2114 under the modified NRC-III rate SLR scenario and an increase of 1.29 m under the modified NRC-III rate (USACE High) SLR scenario. However, our study advances the USACE study in several ways. First, NM14 presented the flood levels at the

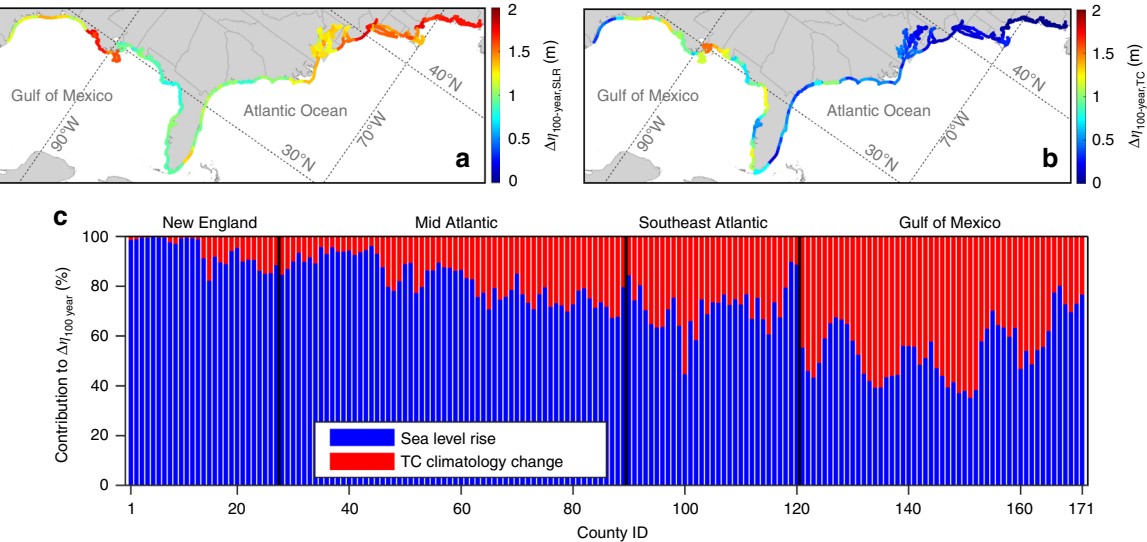

**Fig. 4** Projected contributions of SLR and TC climatology change to changes in the 100-year flood level. **a** Changes in $\eta_{100\text{ year}}$ for the future period of 2070–2095 due to SLR ($\Delta\eta_{100\text{ year, SLR}}$). **b** Changes in $\eta_{100\text{ year}}$ for the future period of 2070–2095 due to TC climatology change ($\Delta\eta_{100\text{ year, TC}}$). **c** Relative contributions of SLR and TC climatology change to the projected changes in $\eta_{100\text{ year}}$. Source data are provided as a Source data file

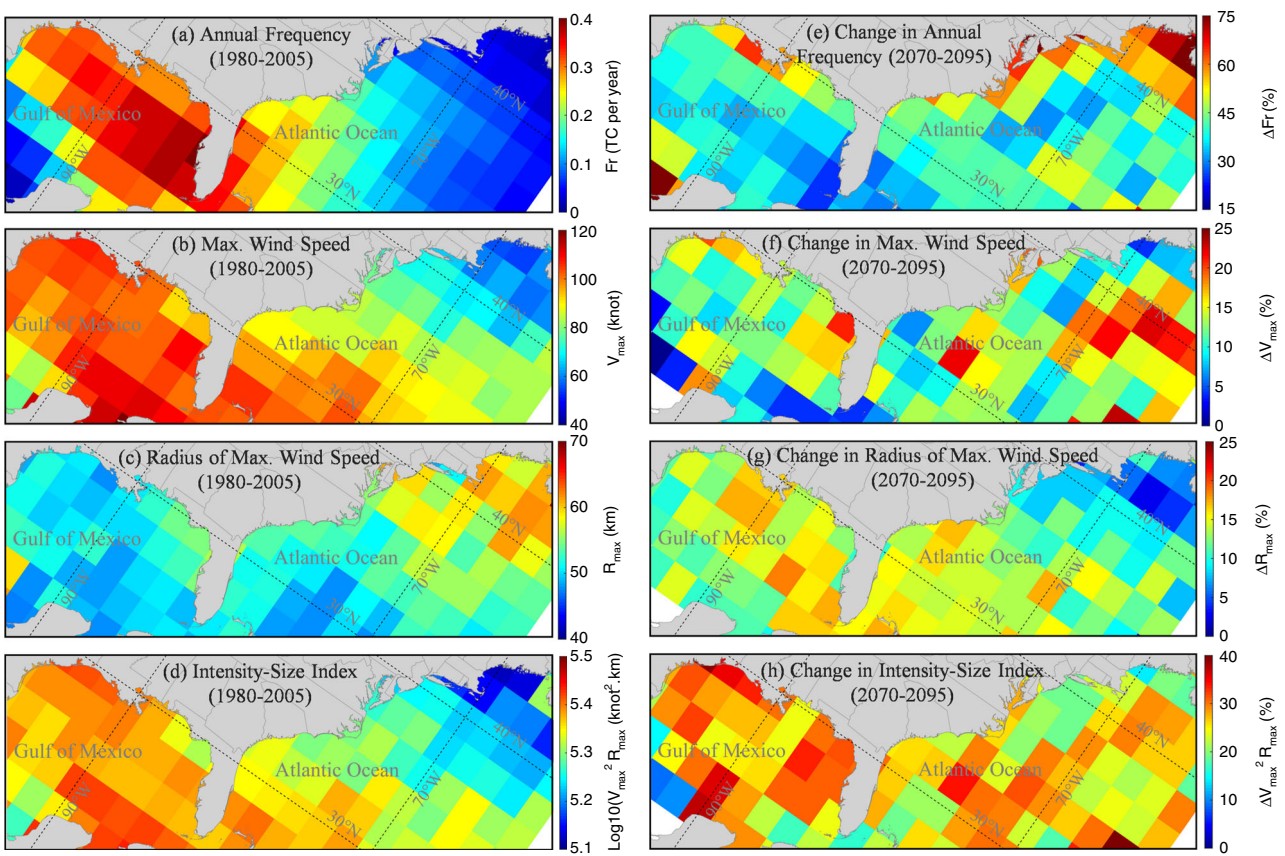

**Fig. 5** Projected changes in TC characteristics from the historical period of 1980–2005 to the future period of 2070–2095. **a–d** NCEP-based TC annual frequency (Fr), maximum wind speed ($V_{max}$), radius of maximum wind speed ($R_{max}$), and the intensity size index $V_{max}^2 R_{max}$ for the historical period. **e–h** Projected changes (weighted average over climate models) in Fr, $V_{max}$, $R_{max}$, and $V_{max}^2 R_{max}$ for the future period. A basin wide weighting factor (which is the average of weighting factors for storm tides over all coastal counties) is used for each climate model. Values represent 90th percentile of TCs (with $V_{max} > 40$ knot) passing through 2 × 2 degrees boxes. Source data are provided as a Source data file

location of 23 tide gauge stations in the Northeastern US whereas we presented the spatially continuous distribution of flood levels along the entire US Atlantic and Gulf Coasts. Second, while NM14 considered deterministic SLR projections, our study accounted for probabilistic SLR projections. Third, NM14 assessed only the effects of SLR on flood levels while we evaluated effects of both SLR and TC climatology change.

Current flood risk mapping from the US Federal Emergency Management Agency (FEMA) has not accounted for the effects of climate change. We found that under the effects of SLR and TC climatology change, the historical 100-year flood level would occur annually in the New England and mid-Atlantic regions and every 1–30 years in the southeast Atlantic and Gulf of Mexico regions in the late 21st century. Thus, we strongly suggest that future flood mapping and flood mitigation planning account for the effects of SLR and TC climatology change.

The basin-scale approach adopted in this study allowed us to evaluate the spatial variation of flood return levels at the county level. Regional-scale (and local-scale) studies and flood mapping require a more detailed variation of flood levels along the coastlines to support regional flood mitigation strategies. Basin-scale studies reveal whether a regional-scale study is essential for a specific region (and how urgent it is), prioritizing regional-scale flood hazard and mitigation studies. Regional scale studies cover a smaller area and thus can have a higher resolution than basin-scale studies. Such studies may also account for elements that are usually missing in larger scale studies such as inundation over coastal floodplains (see further discussion in "Methods" section). Thus, regional-scale studies may be informed by the basin-scale studies, and the results of regional-scale studies in turn may be used to evaluate the accuracy of basin-scale studies. Another benefit of a basin-scale flood hazard study such as presented here is that it reveals regions where SLR, storm climatology change, or both play a role in future changes in flood hazards. Regional-scale studies for regions where the storm climatology change does not impact future flood hazards (e.g. 100-year flood level in Gulf of Maine) may focus on the effect of SLR, while regional-scale studies for Gulf of Mexico should consider the compound effects of SLR and storm climatology change.

We recommend that future studies on coastal flood hazards take into account effects of ETCs and precipitation. In the northeast region of the United States, especially in New England, coastal flooding induced by ETCs are more frequent (but less destructive) than TC-induced flooding[31]. Although ETCs may contribute insignificantly to the 100-year flood levels focused here, they should be accounted for in assessing less extreme flood levels[32]. Also, a recent study has found that although the effect of climate change on ETC storm surges is relatively small on average, large uncertainties exist among climate models[33]. In addition, climate impact studies have shown that climate change will likely increase TC rainfall rates in the future[34,35], which can increase coastal flood hazards for certain regions. Thus, future studies should evaluate changes in coastal flood levels under compound effects of SLR and climatology change of TCs, ETCs, and related precipitation.

## Methods

**Hurricane model.** We use the statistical/deterministic hurricane model developed by refs. [19] to generate large numbers of synthetic storms for different climates. This model uses both thermodynamic and kinematic statistics derived from observations or a climate model to produce synthetic TCs. The model then randomly seeds the basin with weak, warm–core vortices. The motion of vortices is then determined by a beta–and–advection model which uses synthetic environmental wind time series. The wind time series have the monthly means, variances, and covariances calculated from daily data from observations or a climate model and obey a geostrophic turbulence power–law distribution of kinetic energy. The pressure deficit, maximum wind speed, and radius of maximum wind of each storm is calculated using the deterministic coupled air–sea model, Coupled Hurricane Intensity Prediction System (CHIPS)[36].

**Hydrodynamic model.** We use ADCIRC, originally developed by refs. [23,24], to simulate storm tides. We adopt the basin scale computational mesh developed by ref. [25]. The mesh covers the western North Atlantic Ocean, extending between latitudes 8°N and 46°N and longitudes 98°W and 60°W. Details of the hydrodynamic model parameters can be found in ref. [25]. The simulation is driven by wind, pressure, and tidal forcing. We use the analytical model of ref. [37] to calculate the (1 min) storm wind at the gradient level based on the maximum wind speed and radius of maximum wind. For use in the ADCIRC model, we convert the gradient wind to the surface level (10 m above the sea surface) with a velocity reduction factor of 0.85 and an empirical expression of inflow angles[38,39]. We add to the storm wind the surface environmental wind estimated as a fraction (0.55, rotated counterclockwise by 20º) of the storm translation velocity, to account for the asymmetry of the wind field[40]. Finally, we adjust the 1 min wind to a 10 min average with a reduction factor of 0.893[41]. We use the parametric model of ref. [42] to calculate the radial profile of pressure given the pressure deficit. The water level at the open boundaries is specified by eight major tidal constituents K1, K2, M2, N2, O1, P1, Q1, and S2. Tidal data, including amplitudes and phases, are obtained from the global model of ocean tides TPXO8–ATLAS with a 1/30° resolution[43]. The timing of astronomical tide is matched to that of the synthetic TCs.

The hydrodynamic modeling framework was validated using historical TCs and showed satisfactory agreements between measured and modeled storm tides with an overall root-mean-square error, bias, and Willmott skill[44] of 0.31 m, −0.04 m, and 0.9, respectively[25]. In this study, the hydrodynamic modeling framework is driven by synthetic TCs (and the evaluation is shown in Fig. S1). The simulation starts when a storm enters the domain of interest and ends when it leaves the domain (and does not return). The domain of interest expands 800-km seaward of the US East and Gulf Coasts and 300-km landward.

**SLR dataset.** We use the probabilistic, localized SLR projections from ref. [2]. The projections consider ice sheet components (the Greenland, West Antarctic, and the East Antarctic ice sheets); glacier and ice cap surface mass balance; global mean thermal expansion and regional ocean steric and ocean dynamic effects; land water storage; and long term, local, nonclimatic sea level change due to factors such as glacial isostatic adjustment and subsidence. The database provides projections of the probability distribution function (PDF) of SLR at tide gauge stations around the world, under various emission scenarios. In this study, we apply the projection under RCP 8.5 emission scenario for tidal gauges located along the US Atlantic and Gulf Coasts.

**Statistical analysis.** Statistical analysis is performed on the storm tides for each coastal county. Assuming that the storms arrive as a stationary Poisson process under a given climate, the return period of TC-induced storm tide $\eta_{TC}$ exceeding a given level $h$ is[45]

$$T_{\eta_{TC}}(h) = \frac{1}{Fr(1 - P\{\eta_{TC} \le h\})} \quad (1)$$

where $P\{\eta_{TC} \le h\}$ is the cumulative probability distribution (CDF) of peak storm tide and $Fr$ is the TC annual frequency. Previous studies showed that the CDF of TC storm tide is characterized by a long tail and the probability of events representing this tail can be estimated based on the extreme value theory[21,46]. Here we model the tail of the storm tide CDF using the Peaks-Over-Threshold method with a Generalized Pareto Distribution and maximum likelihood estimation[47]. Non-parametric density estimations are used to model the rest of the distribution. We determine a storm tide threshold value to separate the tail from the rest of the distribution. The threshold value is determined by trial and error so that the smallest error in the distribution fitted to the tail is obtained.

The return period of flood level $\eta$ (combination of TC storm tide and SLR) exceeding a given level $h$ is defined as

$$T_\eta(h) = \frac{1}{Fr(1 - P\{\eta \le h\})} \quad (2)$$

where $P\{\eta \le h\}$ is the CDF of flood level, which like refs. [45,48] is calculated through a convolution of the CDF of storm tide and the PDF of relative sea level (RSL):

$$P\{\eta \le h\} = P\{\eta_{TC} + \eta_{RSL} \le h\} = \int_{-\infty}^{+\infty} P\{\eta_{TC} \le h - x\} f_{\eta_{RSL}}(x) dx \quad (3)$$

where $\eta_{RSL}$ is the RSL, which represents the mean sea level in any year relative to the mean sea level in the baseline year 2000, and $f_{\eta_{RSL}}(x)$ is the PDF of RSL.

We note that projections from climate models may be biased. Therefore, we bias correct the GCM-projected storm tide climatology before combining it with RSL distributions to estimate flood hazards. Similar to refs. [45,48], we bias correct the storm frequency and storm tide CDF for each GCM by comparing the GCM-estimated frequencies and CDFs for the historical period with NCEP-based estimates and assuming that the biases calculated for the historical period can be employed to bias correct future projections. In particular, we bias correct the storm tide CDF through quantile–quantile mapping. For each return period (with an

increment step of 1 year), we calculated the bias by subtracting the NCEP-based return level from the climate-model-estimated return level for the historical period and subtracted this bias from the climate-model-projected return level for the future period.

In addition to the future projection of storm tides from each climate model, we also calculate a single composite projection which represents the weighted–average value over all climate models. The weighting factor assigned to each climate model is determined by comparing the NCEP-based storm tide return levels with those projected by the climate models for the historical period. Specifically, the weighting factor $W_i$ of the climate model $i$ is simply calculated as $W_i = S_i/\sum S_i$, where $S_i$ is the Wilmott skill score ($0 < S < 1$, $S = 1$ means a perfect fit) for estimating the storm tide return levels.

**Limitations.** There are some limitations to the results presented in this paper. The hydrodynamic model neglects the wave setup (i.e., the water level increase at the coast due to breaking waves in the surf zone). Wave setup can increase the flood levels up to a few tens of centimetres[25,49,50]. The wave setup can be computed by coupling the hydrodynamic model with a spectral wave model. However, the computational cost would increase significantly (note that we simulated about 65,000 synthetic TCs). Another limitation of the hydrodynamic model is the spatial resolution of the computational mesh, which is about 1 km along the coastlines. A higher resolution mesh could reduce numerical errors and better resolve physical processes especially the wave setup (when the hydrodynamic model is coupled with a wave model as in ref. [25]). The basin-scale computational mesh does not cover coastal floodplains. Including the floodplains requires a higher resolution mesh that resolves the complex features of coastal areas especially in urbanized regions (e.g., flood protection systems, roads, narrow waterways, etc.). These components may be incorporated in future assessments when computational resources allow.

We assume that SLR and storm tides are independent and, thus, the nonlinear interactions are neglected. Although the nonlinear effects have been shown to be negligible for some coastal areas[21,51], depending on the bathymetry and geometry of the coast, SLR could influence the tidal range and storm surge height[52–54]. Accounting for this nonlinear interaction through direct simulation is computationally expensive, if possible, for full probabilistic assessments, as it would require simulating numerous combinations of all possible storms and SLR scenarios. (Direct simulations may be applied for studies with substantially reduced number of scenarios, for example, through focusing on a few selected SLR scenarios[54]). Developing parametric approximations may be necessary. Ultimately, SLR and storm climatology changes are correlated in the climate and ocean system[55]. Here they are assumed independent conditioned on the emission scenario (RCP 8.5) and the overall climate modeling (CMIP5). An integrated modeling framework that can estimate SLR and storm activity together for each climate projection may help both accounting for their correlation more accurately and providing reduced sample sizes that can be directly simulated to fully account for the nonlinear interactions between SLR and storm tides.

## Data availability
The source data underlying all figures (Figs. 1–5 and Supplementary Figs. 1–8) are provided as a Source Data file. All data are available from the authors upon reasonable request.

## Code availability
Codes used for this work are available from the authors upon reasonable request.

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

## Acknowledgements

This material is based upon work supported by the National Science Foundation (EAR-1520683) and the Princeton Environmental Institute at Princeton University. The first author performed the numerical simulations and statistical analysis when he was formerly an associate research scholar in the Department of Civil and Environmental Engineering at Princeton University.

## Author contributions

R.M. performed hydrodynamic modeling of storm tides and conducted statistical analysis. N.L. designed the project and supervised storm tide simulations and statistical analysis. K.E. carried out numerical modeling of synthetic tropical cyclones. K.F. assisted in the statistical analysis. R.M. and N.L. wrote the paper and K.E. edited the paper. All authors participated in technical discussions.

## Additional information

**Competing interests:** The authors declare no competing interests.

