## [Peer Review File · Nature Communications]

Reviewers' comments:

Reviewer #1 (Remarks to the Author):

The paper "Climate Change Exacerbates Hurricane Flood Hazards Along U.S. Atlantic Coast in Spatially Varying Patterns" by Marsooli et al. investigates end-of-century impacts of changes in sea level rise and tropical cyclone characteristics on hurricane flood hazard along the U.S. Atlantic and Gulf coast. The paper uses a set of global climate models and creates synthetic hurricanes, which's wind fields drive a hydrodynamic model that allows assessing hurricane storm surge levels at various return levels. The authors show substantial increases in future storm surges and attribute those changes majorly to sea level rise in the New England and mid-Atlantic region and to a mix of sea level rise and changes in the hurricane climatology in the south-Atlantic and Gulf coast. The study presents novel results that have high societal implications and is worth publishing in Nature Communications. The paper is well structured, has high-quality images, and is well written. I have some mostly minor comments that are described below.

Best regards,

Andreas Prein

General Comments

- 1) As far as I understand from your paper you assume that changes in sea level rise and changes in the TC climatology are additive. How good of an assumption is that? E.g., could sea level rise induced changes in the coastline have a non-linear effect on TC storm surge?
- 2) You describe that you use sea level rise (SLR) projections from Kopp et al. (2014), which are probabilistic, but in your paper, it seems that you use a deterministic value for (SLR) while you account for uncertainties in TC climatology by including 6 GCMs in your study. Accounting for, or at least discussing the impact of SLR uncertainties on your results seems to be important due to the large uncertainty in this variable.
- 3) Are TCs the major contributor to coastal flooding in New England or are other storms (e.g., extratropical cyclones) also important? I guess my main question is if there could be changes in other atmospheric processes that could lead to an increase in storm surge in New-England. You briefly discuss this in the Appendix but I think it is important to mention in the main article.
- 4) One component of TC flooding that you did not mention in your study is flooding from TC precipitation. There are many studies that suggest that TC precipitation is increasing due to climate

change (e.g. Bacmeister et al. 2018, Gutmann et al. 2018), which could further increase the flood hazard in addition to SLR and TC climatology changes.

5) Did you see any evidence for changes in TC speed or size in your future climate simulations? Changes in these two characteristics would certainly have an impact on TC storm surge.

6) Are the synthetic TC tracks from CMIP5 models comparable to observed tracks (e.g., spatial distribution, intensity)?

Specific comments:

Title: Why are you only focusing on the Atlantic coast in your title and do not mention the Gulf coast?

L41: a large spectra

L70-6: Why did you choose these models? Do they span the uncertainty range of the CMIP5 ensemble and have a good performance in the current climate?

L86: You mention that you use quantile-quantile mapping for the bias correction. Does this method allow future extreme values that are larger than the extremes in the current climate period? This is a typical limitation of many bias correction approaches.

L106: I struggled with the word “satisfactory” here since comparing your modeled data with the observation is really hard due to the large uncertainties in the observational return periods in Fig. S1. Maybe acceptable would be a better word.

L221: “...0.49 m (16%)...”

Figure 5: Adding labels to the figure panels would help the reader to easier find the figures that you are referring to in the text. Additionally, I would suggest adding hatching in areas where the changes are significant in the climate change plots.

L330: You use the acronym RSL here for the first time. Should this be SLR (sea level rise)? If yes, please make sure to replace RSL throughout your manuscript with SLR.

Literature:

Bacmeister, J.T., Reed, K.A., Hannay, C., Lawrence, P., Bates, S., Truesdale, J.E., Rosenbloom, N. and Levy, M., 2018. Projected changes in tropical cyclone activity under future warming scenarios using a high-resolution climate model. *Climatic Change*, 146(3-4), pp.547-560.

Gutmann, E.D., Rasmussen, R.M., Liu, C., Ikeda, K., Bruyere, C.L., Done, J.M., Garrè, L., Friis-Hansen, P. and Veldore, V., 2018. Changes in hurricanes from a 13-yr convection-permitting pseudo-global warming simulation. *Journal of Climate*, 31(9), pp.3643-3657.

Kopp, R.E., Horton, R.M., Little, C.M., Mitrovica, J.X., Oppenheimer, M., Rasmussen, D.J., Strauss, B.H. and Tebaldi, C., 2014. Probabilistic 21st and 22nd century sea-level projections at a global network of tide-gauge sites. *Earth's future*, 2(8), pp.383-406.

Kossin, J.P., 2018. A global slowdown of tropical-cyclone translation speed. *Nature*, 558(7708), p.104.

Reviewer #2 (Remarks to the Author):

This study presents an evaluation of the combined impacts of sea-level rise and tropical cyclone climatology under climate change on coastal flooding along the United States East Coast and the Gulf of Mexico. The study is based on a coupled climatology and hydrodynamic modeling framework. Climate change impacts on tropical cyclone climatology are based on CMIP5 global climate models and a statistic-deterministic model used to generate a large numbers of synthetic storms. A widely used hydrodynamic model (ADCIRC) is used to simulate storm surge along the coast. Probabilistic predictions of future sea-level rise are linearly added to the simulated water levels along the coast. A very informative and interesting spatial analyses of the relevance of SLR and TC climatology to the frequency of occurrence of extreme events along the coast is presented. By focusing on the 1% chance events, it brings useful information regarding the changes to the future flood risks along the coast. Figure 5 is especially informative by clearly demonstrating the importance of each component to the total coastal flood levels in the different regions of the country. It not only demonstrates the importance of SLR and climate change impacts on TC for coastal flooding but also spatially quantifies its importance and magnitude. The amount of simulations performed is not trivial. The methodology seems to be well validated based on historical records and the NCEP reanalyzes comparison. ADCIRC is a well validated and widely used model in the scientific community, and commonly used for this type of studies. By demonstrating the predicted changes in frequency for coastal flooding throughout the East Coast and the GOM, the study brings a refreshing reminder of the increasing risks that coastal communities will face in the near future.

I have some minor suggestions for the authors:

In the Summary:

For the sentence: "Under compound effects of SLR and TC climatology change, the historical 21 100-year flood level would occur annually in the New England and mid-Atlantic regions and 22 every 1-30 years in the southeast Atlantic and Gulf of Mexico regions."

When and under what scenario? Is it (2070- 2095)? And (under the emission scenario RCP 8.5) ?

Methods:

How is the timing effects of astronomical tides incorporated? For example, the tides are time dependent and the hypothetical storms might not be. While tidal range in the GOM is small, that is not the case in Maine. How is that variation impacting peak surges and its implications on the frequency analyses?

How this study does compare to previous studies? For example, the North Atlantic Coast Comprehensive Study Phase I: statistical analysis of historical extreme water levels with sea level change. In the discussion, the authors mentioned FEMA studies but did not mentioned the latest USACE studies. Did they have similar results? Different approaches? How does this study advances the USACE studies?

Ref: Nadal-Caraballo, Norberto C.; Melby, Jeffrey A. (Coastal and Hydraulics Laboratory (U.S.) Engineer Research and Development Center (U.S.), 2014-09)

Line 121: I do not see the physical meaning or the local implication of the average of flood levels between counties. I can see, however, how the average change can give an overall idea of the expected magnitudes of change.

Line 132 – what is very-likely? I see this in Figure 2 caption: s (i.e. 90% statistical confidence interval).

Nice discussion on the limitations of the study and I certainly agree that these aspects are important and should be revisited in future studies.

Figure 2 is very informative. There seems to be an almost linear pattern of increase between the historical (black) and compound (red) lines for the cases where the TC climatology is not significant. Is that really the case for places where SLR is the predominant signal? Can we provide some guidance to coastal communities here? For example, can we expect a linear increase on locations where SLR is the prevalent signal (i.e., you can achieve the red line by simply adding a given SLR interval to the black line - up to a certain level of return period?). In contrast, where TC climatology changes becomes more significant, that signal needs to be taken into consideration (e.g., Gulf of Mexico and Southern Atlantic) for any coastal risk future prediction.

Figure 4 is great.

Celso Ferreira, Ph.D., P.E.

Reviewer #3 (Remarks to the Author):

Manuscript: Climate Change Exacerbates Hurricane Flood Hazards Along U.S. Atlantic Coast in Spatially Varying Patterns

The manuscript presents a climatology-hydrodynamic modeling and statistical-analysis approach to examine future (late 21st century) flood hazards along the US/Gulf coast. I found the manuscript an interesting read – and it is generally well-written. The data and methods are sound and supported by previously published works.

I have one major comment (below) and a few minor comments.

Major Comments

I believe that studies, such as the one presented in this manuscript, that focus on future flood hazards (under climate change including SLR and TC climatology changes) for large areas of coastline are important. However, I often find that there is a large gap in such broad results and how it will eventually be used in practice – for mitigation measures and decisions to be made. Can you comment on how using a single (lat/lon) point for an entire coastal county is a reasonable assumption for your approach? Can you also include additional discussion on how the presented method/results may compare to regional studies that are able to focus on finer details by using higher-resolution meshes and wetting/drying of the coastal floodplain, etc.? How can this study then inform more regionally-focused studies? How could regional studies inform a study for larger-regions, such as this one? I think such questions are important to consider in the revision of this article.

Minor Comments

Line 25: Should this not be: “Coastal flooding responds to the impacts of climate change, including sea level rise (SLR).”? SLR is an outcome of climate change and the two are not necessarily separated. However, one could make an argument that relative sea level rise could be separate when considering local subsidence.

Line 58: It looks like you also do the U.S. Gulf Coast as well (Line 91, for example). Make that clear in the manuscript.

Lines 354-357: A higher resolution mesh that covers the coastal floodplain will also likely result in changes to the results. I suggest briefly commenting on this.

Line 357-361: I suggest citing additional articles here that show the nonlinearities of SLR-surge are important, and that those that (2004) the dynamics of sea level rise and coastal flooding on a changing landscape." *Geophysical Research Letters* 41(3): 927-934.

- Bilskie, M. V., et al. (2014). "Dynamics of sea level rise and coastal flooding on a changing landscape." *Geophysical Research Letters* 41(3): 927-934.
- Bilskie, M. V., et al. (2019). "Development of Return Period Stillwater Floodplains for the Northern Gulf of Mexico under the Coastal Dynamics of Sea Level Rise." *J. Waterway, Port, Coastal, Ocean Eng.* 145(2): 04018043.

- Atkinson, J. H., et al. (2013). "Sea-level rise effects on storm surge and nearshore waves on the Texas coast: influence of landscape and storm characteristics." *Journal of Waterway, Port, Coastal, and Ocean Engineering* 139(2): 98-117.

Reviewer #1 (Remarks to the Author):

The paper “Climate Change Exacerbates Hurricane Flood Hazards Along U.S. Atlantic Coast in Spatially Varying Patterns” by Marsooli et al. investigates end-of-century impacts of changes in sea level rise and tropical cyclone characteristics on hurricane flood hazard along the U.S. Atlantic and Gulf coast. The paper uses a set of global climate models and creates synthetic hurricanes, which’s wind fields drive a hydrodynamic model that allows assessing hurricane storm surge levels at various return levels. The authors show substantial increases in future storm surges and attribute those changes majorly to sea level rise in the New England and mid-Atlantic region and to a mix of sea level rise and changes in the hurricane climatology in the south-Atlantic and Gulf coast. The study presents novel results that have high societal implications and is worth publishing in Nature Communications. The paper is well structured, has high-quality images, and is well written. I have some mostly minor comments that are described below.

Best regards,
Andreas Prein

General Comments

1) As far as I understand from your paper you assume that changes in sea level rise and changes in the TC climatology are additive. How good of an assumption is that? E.g., could sea level rise induced changes in the coastline have a non-linear effect on TC storm surge?

Non-linear interactions between sea level rise and TC storm tide depend on the bathymetry and shoreline geometry and, thus, can vary from one region to another region. While some studies showed that nonlinear interactions can impact tides and storm surges (e.g. Atkinson et al. 2013; Bilskie et al. 2014), other studies showed that the nonlinear effects are negligible (e.g. Lin et al. 2012 and Orton et al. 2015).

To account for the nonlinear interactions, one ideally should carry out each TC storm tide simulation (in this study, ~ 65,000 simulations) under each scenario of SLR. In studies that consider probabilistic SLR projections, such as the present study, one needs to perform each TC simulation under large numbers of SLR scenarios, which is currently impractical. In climate impact studies, the resulting error due to the neglected nonlinear interactions is outweighed by other sources of uncertainty such as projections of future climate conditions. We have added to the last paragraph of section “Limitation” in Methods a brief discussion about the non-linear interaction between sea level rise and storm tide.

Lin, N., Emanuel, K., Oppenheimer, M. & Vanmarcke, E. Physically based assessment of hurricane surge threat under climate change. *Nat. Clim. Chang.* 2, 462–467 (2012).

Orton, P. et al. New York City Panel on Climate Change 2015 Report Chapter 4: Dynamic Coastal Flood Modeling. *Ann. N. Y. Acad. Sci.* 1336, 56–66 (2015).

Atkinson, J., McKee Smith, J. & Bender, C. Sea-Level Rise Effects on Storm Surge and Nearshore Waves on the Texas Coast: Influence of Landscape and Storm Characteristics. *J. Waterw. Port, Coastal, Ocean Eng.* 139, 98-117 (2013).

Bilskie, M. V, Hagen, S. C., Medeiros, S. C. & Passeri, D. L. Dynamics of sea level rise and coastal flooding on a changing landscape. 927-934 (2014). doi:10.1002/2013GL058759.

2) You describe that you use sea level rise (SLR) projections from Kopp et al. (2014), which are probabilistic, but in your paper, it seems that you use a deterministic value for (SLR) while you account for uncertainties in TC climatology by including 6 GCMs in your study. Accounting for, or at least discussing the impact of SLR uncertainties on your results seems to be important due to the large uncertainty in this variable.

Response: The present study is not based on a deterministic value for SLR. We used probabilistic projections of SLR. The return periods are based on a convolution of cumulative distribution function of storm tide and the probability distribution function of relative sea level. Please see “statistical analysis” in section “methods”. To make it more clear, we have changed some wording. E.g., in “Modeling and Analysis”, rather than saying “Combining storm tide and SLR datasets, ...”, we have changed it to “Combining the probabilistic projections of storm tide and SLR, we perform statistical analysis to estimate the return periods of flood heights (see Methods).”.

3) Are TCs the major contributor to coastal flooding in New England or are other storms (e.g., extratropical cyclones) also important? I guess my main question is if there could be changes in other atmospheric processes that could lead to an increase in storm surge in New-England. You briefly discuss this in the Appendix but I think it is important to mention in the main article.

Response: In the northeastern U.S., extratropical cyclones (ETCs) are more frequent than tropical cyclones (TCs). On the other hand, TCs have generated the most devastating flood events in the region. For example, among the most destructive storms in New England are the 1938 New England Hurricane, known as Yankee Clipper, Hurricanes Carol and Edna in 1954, Hurricane Bob in 1991, and Hurricane Sandy in 2012.

In a separate study, we have recently evaluated the effects of climate change on ETC storm surges (Lin et al. 2019). In this recent study, we found small to moderate effects of climate change on ETC storm surges. We have added new texts to the revised manuscript to address the reviewer’s comment. Please see the fourth paragraph of section “Discussion” in the revised manuscript.

Lin, N., R. Marsooli, B.A. Colle. 2019. Storm Surge Return Levels Induced by the Mid-to-Late-21st-Century Extratropical Cyclones in the Northeastern United States. *Climate Change*. <https://doi.org/10.1007/s10584-019-02431-8>

4) One component of TC flooding that you did not mention in your study is flooding from TC precipitation. There are many studies that suggest that TC precipitation is increasing due to

climate change (e.g, Bacmeister et al. 2018, Gutmann et al. 2018), which could further increase the flood hazard in addition to SLR and TC climatology changes.

Response: Thanks for raising this great point. We agree that changes to TC precipitation may impact coastal flood hazards, but this compound effect is not well understood yet. We have added a comment to the fourth paragraph of section “Discussion” in the revised manuscript. TC rainfall induced inland flooding is another important topic, which is out of the scope of this study.

5) Did you see any evidence for changes in TC speed or size in your future climate simulations? Changes in these two characteristics would certainly have an impact on TC storm surge.

Response: We found that the size of TCs off the entire U.S. East and Gulf Coasts increases (up to 21% increase) from the historical period to the future period. Figure 5 in the manuscript displays changes in the TC size.

An analysis of the synthetic TC speed indicates that the number of slow-moving TCs will increase by the end of 21st century (under RCP8.5). For example, the density histograms shown below indicate an increase in the number of slow-moving TCs and a decrease in the number of fast-moving TCs. Slower TCs can worsen coastal flooding by dumping more rain in a specific area, which can lead to catastrophic inland flooding as exemplified by Hurricane Harvey (2017). Moreover, a slow-moving TC allows winds to blow onshore for a longer period, resulting in a larger and longer storm surge event. We have added a brief discussion of the effects of slow-moving TCs on coastal flood hazards. Please see the last few sentences in the third paragraph of section “Relative Impact of SLR and TC Climatology Change”.

Figure 1. Histograms of TC translation speed (in m/s) for high-latitude (top panels; lat>36 degree) and low-latitude (bottom panels; lat<36 degree) regions in our study area. Black bars represent the historical period of 1980–2005 and red bars represent the future period of 2070–2095. The vertical axis represents the density, i.e. normalized number of TCs (normalized by the total number of TCs in the dataset).

6) Are the synthetic TC tracks from CMIP5 models comparable to observed tracks (e.g., spatial distribution, intensity)?

Response: Please see “Emanuel, K., R. Sundararajan, and J. Williams, 2008: Hurricanes and global warming: Results from downscaling IPCC AR4 simulations. *Bull. Amer. Meteor. Soc.*, **89**, 347-367.” where a fairly detailed comparison is made with observations. The quality of the comparison varies with model, of course, and there can be large differences among results downscaled from different models. As with any regional climate metric, it is best to use a diversity of climate models, as we do here. Also, to address the uncertainty in the climate model projections, we have applied bias correction based on comparing the climate-model generated and reanalysis-driven storm surge statistics.

Specific comments:

Title: Why are you only focusing on the Atlantic coast in your title and do not mention the Gulf coast?

Response: Good point! The title of the revised manuscript is edited to read as “Climate Change Exacerbates Hurricane Flood Hazards Along U.S. Atlantic and Gulf Coasts in Spatially Varying Patterns”. We also added “Gulf Coast” to the main text accordingly.

L41: a large spectra

Response: We meant to use the plural form of spectrum.

L70-6: Why did you choose these models? Do they span the uncertainty range of the CMIP5 ensemble and have a good performance in the current climate?

Response: The selected models are those that recorded climate variables required for generating the synthetic TCs by the hurricane model, and they were also used in previous studies. They do provide a wide range of predictions, but they may not fully cover the uncertainty range of the CMIP5 models. We have added to the revised manuscript “Given data availability and following previous studies, we consider projections from six CIMP5 models:...” Most other CMPI5 models do not provide sufficient input for our approach. It is also computationally difficult to test all CMIP5 models, given that we need to generate at least thousands of storms and storm tide scenarios for each model. To account for the potential bias and uncertainties, we made efforts to make bias corrections and weighted model combinations.

L86: You mention that you use quantile-quantile mapping for the bias correction. Does this method allow future extreme values that are larger than the extremes in the current climate period? This is a typical limitation of many bias correction approaches.

Response: We calculated biases not based on empirical points but based on the fitted return period curves. For each return period (e.g. 1000-year), we calculated the bias by subtracting the historical NCEP-based return level corresponding with that return period from the historical model-based return level corresponding with the same return period. This approach allowed us to calculate the bias for each return periods (with an increment step of one year). This approach thus allows the future extreme values to be larger than current extremes. This is now clarified in the revised manuscript. Please see section “Statistical analysis” in the revised manuscript.

L106: I struggled with the word “satisfactory” here since comparing your modeled data with the observation is really hard due to the large uncertainties in the observational return periods in Fig. S1. Maybe acceptable would be a better word.

Response: “satisfactory” is replaced with “good” in the revised manuscript.

L221: “...0.49 m (16%)...”

Response: Corrected.

Figure 5: Adding labels to the figure panels would help the reader to easier find the figures that you are referring to in the text. Additionally, I would suggest adding hatching in areas where the changes are significant in the climate change plots.

Response: We tried to hatch the figure but it becomes too busy and some labels become difficult to read. However, we added a title to each panel which helps the reader to easier find the figures that are discussed in the text. Please see Figure 5 in the revised manuscript.

L330: You use the acronym RSL here for the first time. Should this be SLR (sea level rise)? If yes, please make sure to replace RSL throughout your manuscript with SLR.

Response: RSL stands for relative sea level, which represents the mean sea level in any given year compared to the mean sea level in the baseline year 2000. The definition of RSL is now added to the revised manuscript. Please see section “Statistical analysis” in the revised manuscript.

Literature:

- Bacmeister, J.T., Reed, K.A., Hannay, C., Lawrence, P., Bates, S., Truesdale, J.E., Rosenbloom, N. and Levy, M., 2018. Projected changes in tropical cyclone activity under future warming scenarios using a high-resolution climate model. *Climatic Change*, 146(3-4), pp.547-560.
- Gutmann, E.D., Rasmussen, R.M., Liu, C., Ikeda, K., Bruyere, C.L., Done, J.M., Garrè, L., Friis-Hansen, P. and Veldore, V., 2018. Changes in hurricanes from a 13-yr convection-permitting pseudo-global warming simulation. *Journal of Climate*, 31(9), pp.3643-3657.
- Kopp, R.E., Horton, R.M., Little, C.M., Mitrovica, J.X., Oppenheimer, M., Rasmussen, D.J., Strauss, B.H. and Tebaldi, C., 2014. Probabilistic 21st and 22nd century sea level projections at a global network of tide gauge sites. *Earth's future*, 2(8), pp.383-406.
- Kossin, J.P., 2018. A global slowdown of tropical-cyclone translation speed. *Nature*, 558(7708), p.104.

Reviewer #2 (Remarks to the Author):

This study presents an evaluation of the combined impacts of sea-level rise and tropical cyclone climatology under climate change on coastal flooding along the United States East Coast and the Gulf of Mexico. The study is based on a coupled climatology and hydrodynamic modeling

framework. Climate change impacts on tropical cyclone climatology are based on CMIP5 global climate models and a statistic-deterministic model used to generate a large numbers of synthetic storms. A widely used hydrodynamic model (ADCIRC) is used to simulate storm surge along the coast. Probabilistic predictions of future sea-level rise are linearly added to the simulated water levels along the coast. A very informative and interesting spatial analyses of the relevance of SLR and TC climatology to the frequency of occurrence of extreme events along the coast is presented. By focusing on the 1% chance events, it brings useful information regarding the changes to the future flood risks along the coast. Figure 5 is especially informative by clearly demonstrating the importance of each component to the total coastal flood levels in the different regions of the country. It not only demonstrates the importance of SLR and climate change impacts on TC for coastal flooding but also spatially quantifies its importance and magnitude. The amount of simulations performed is not trivial. The methodology seems to be well validated based on historical records and the NCEP reanalyzes comparison. ADCIRC is a well validated and widely used model in the scientific community, and commonly used for this type of studies. By demonstrating the predicted changes in frequency for coastal flooding throughout the East Coast and the GOM, the study brings a refreshing reminder of the increasing risks that coastal communities will face in the near future.

I have some minor suggestions for the authors:

In the Summary:

For the sentence: “Under compound effects of SLR and TC climatology change, the historical 21 100–year flood level would occur annually in the New England and mid–Atlantic regions and 22 every 1–30 years in the southeast Atlantic and Gulf of Mexico regions.”

When and under what scenario? Is it (2070– 2095)? And (under the emission scenario RCP 8.5)?

Response: We thank the reviewer for bringing the missing info to our attention. That’s right, the findings are for the late 21st century (2070-2095) under the RCP8.5 scenario. The missing info is added. Please see the summary in the revised manuscript.

Methods:

How is the timing effects of astronomical tides incorporated? For example, the tides are time dependent and the hypothetical storms might not be. While tidal range in the GOM is small, that is not the case in Maine. How is that variation impacting peak surges and its implications on the frequency analyses?

Response: Hydrodynamic simulations conducted in the present study include both storm surge and astronomical tides, allowing us to directly account for non-linear tide-surge interactions. The timing of tides for a given TC is matched to that of the synthetic TCs. The synthetic TCs do have time (year, day, hour) dependent on the large-scale climate environment. This is now added to the text “The timing of astronomical tide is matched to that of the synthetic TCs (consistent with the climatology).”

How this study does compare to previous studies? For example, the North Atlantic Coast Comprehensive Study Phase I: statistical analysis of historical extreme water levels with sea level change. In the discussion, the authors mentioned FEMA studies but did not mentioned the

latest USACE studies. Did they have similar results? Different approaches? How does this study advance the USACE studies?

Ref: Nadal-Caraballo, Norberto C.; Melby, Jeffrey A. (Coastal and Hydraulics Laboratory (U.S.) Engineer Research and Development Center (U.S.), 2014-09)

Response: The present study advances the USACE study in several ways. First, the USACE study presented the flood levels at the location of 23 tide gauge stations in the Northeastern U.S. whereas we presented the spatially continuous distribution of flood levels along the entire U.S. Atlantic and Gulf Coasts. Second, while the USACE study considered deterministic SLR projections, our study accounted for probabilistic SLR projections. Third, the USACE study assessed only the effects of SLR on flood levels while we evaluated effects of both SLR and TC climatology change.

Our findings on the effects of SLR on flood return levels is comparable to those from the USACE study. We project, for example, that SLR causes an increase of 1.38 m in the 100-year flood level at the New York county from the historical period of 1980-2005 to the future period of 2070-2095 under RCP 8.5 emission scenario. Projections from the USACE study for the New York City (The Battery NY) indicate that SLR causes an increase of 0.8 m in the 100-year flood level from 2014 and 2114 under the “Modified NRC-III rate” SLR scenario and an increase of 1.29 m under the “Modified NRC-III rate (USACE High)” SLR scenario. However, we were not able to perform a direct comparison between the two studies as our SLR scenarios differ from those used in the USACE study.

We have added a new paragraph to discuss how the present study advances the USACE study. Please see the third paragraph in the section “Discussion” of the revised manuscript.

Line 121: I do not see the physical meaning or the local implication of the average of flood levels between counties. I can see, however, how the average change can give an overall idea of the expected magnitudes of change.

Response: We agree that county-level findings give an overall view of expected changes. The results are useful to identify regions that will encounter higher flood levels as a result of not only SLR but also TC climatology change. Future studies should be conducted to better quantify spatial variation of flood hazard changes at a local scale. This is especially needed as flood level is also a function of bathymetry and coastline geometry that can vary in a given region. Our results can be used as a guide to identify regions where the future flood levels could surpass the typical height of existing flood mitigation measures and, thus, assist decision-makers to prioritize regional-scale flood hazard and mitigation studies, as indicated in the first paragraph in Discussion. Also, the added discussion on the USACE study enhance the point, as the USACE study is developed for engineering applications. Indeed, in a separate study (forthcoming), we have used the basin-scale simulation produced here as input to drive a high-resolution mesh focused on Jamaica Bay, New York, to support local planning and flood defense design.

Line 132 – what is very-likely? I see this in Figure 2 caption: s (i.e. 90% statistical confidence interval).

Response: It is 5th-95th percentile (90% confidence interval). The sentence is edited in the revised manuscript.

Nice discussion on the limitations of the study and I certainly agree that these aspects are important and should be revisited in future studies.

Response: Thanks!

Figure 2 is very informative. There seems to be an almost linear pattern of increase between the historical (black) and compound (red) lines for the cases where the TC climatology is not significant. Is that really the case for places where SLR is the predominant signal? Can we provide some guidance to coastal communities here? For example, can we expect a linear increase on locations where SLR is the prevalent signal (i.e., you can achieve the red line by simply adding a given SLR interval to the black line - up to a certain level of return period?). In contrast, where TC climatology changes becomes more significant, that signal needs to be taken into consideration (e.g., Gulf of Mexico and Southern Atlantic) for any coastal risk future prediction.

Response: The return period curves in Figure 2 imply that SLR linearly shifts the historical curves higher. As the reviewer mentioned, this pattern is visible in areas that the effects of TC climatology change are negligible. Although a linear superposition of a given SLR on the historical flood return levels may result in an estimate of future flood levels, we would be careful with recommending such approach for flood mitigation planning in coastal communities. In the present study, we neglected non-linear interactions of SLR with tides and storm surges. Depending on the bathymetry, tidal range, and shoreline geometry, the non-linear interactions could affect flood levels with specific return periods (“up to a certain level of return period”). The topic of the effects of non-linear SLR-tide-surge interactions on flood return levels should be studied in more details. In section “limitations”, we briefly discuss effects of nonlinear interactions on flood levels.

Figure 4 is great.

Celso Ferreira, Ph.D., P.E.

Reviewer #3 (Remarks to the Author):

See attached PDF for comments.

-Matt Bilskie

Manuscript: Climate Change Exacerbates Hurricane Flood Hazards Along U.S. Atlantic Coast in Spatially Varying Patterns

The manuscript presents a climatology-hydrodynamic modeling and statistical-analysis approach to examine future (late 21st century) flood hazards along the US/Gulf coast. I found the

manuscript an interesting read – and it is generally well-written. The data and methods are sound and supported by previously published works.

I have one major comment (below) and a few minor comments.

Major Comments

I believe that studies, such as the one presented in this manuscript, that focus on future flood hazards (under climate change including SLR and TC climatology changes) for large areas of coastline are important. However, I often find that there is a large gap in such broad results and how it will eventually be used in practice – for mitigation measures and decisions to be made. Can you comment on how using a single (lat/lon) point for an entire coastal county is a reasonable assumption for your approach? Can you also include additional discussion on how the presented method/results may compare to regional studies that are able to focus on finer details by using higher-resolution meshes and wetting/drying of the coastal floodplain, etc.? How can this study then inform more regionally-focused studies? How could regional studies inform a study for larger-regions, such as this one? I think such questions are important to consider in the revision of this article.

Response: The county-scale approach adopted in our study allowed us to evaluate the spatial variation of flood levels at a basin scale. This approach is reasonable for basin- and global-scale studies but not recommended in regional-scale studies for planning and implementing flood mitigation strategies. Our results provide first-order estimates of changes in flood hazards for each county. Such basin-scale estimates could inform decision-makers whether a regional-scale study is necessary for a specific region (and how urgent it is). One can use results from a basin-scale study to identify regions where the future flood levels could surpass the typical height of existing flood mitigation measures and, thus, prioritize regional-scale flood hazard and mitigation studies. Regional-scale studies cover a smaller area and thus can have a higher resolution than basin-scale studies. It is also feasible in a regional-scale model to account for elements that are usually missing in larger scale studies (e.g. wetting/drying of floodplains, wave setup, and other elements described in the section “limitation” in the manuscript) which can result in more accurate estimates of flood levels. The results of such regional-scale studies may be used to evaluate the accuracy of basin-scale studies. Indeed, in a separate study (forthcoming), we have used the basin-scale simulation produced here as input to drive a high-resolution mesh focused on Jamaica Bay, New York, to support local planning and flood defense design.

Another benefit of a basin-scale study like the present study is that it reveals regions where sea level rise, storm climatology change, or both play a role in future changes in flood hazards. Regional-scale studies for regions that the storm climatology change does not impact future flood hazards may only focus on the effect of SLR, leading to a reduction in time and efforts required to complete the study. For example, regional-scale flood hazard studies for coastal counties in the states of Maine and New Hemisphere may neglect effects of storm climatology change, as Figure 4 in the manuscript indicates that SLR is the main cause of the increase in future flood levels. In regions that the effects of storm climatology change are small, future studies may neglect such effects but utilize a safety factor that accounts for the neglected effects. The safety factor may be approximated based on the contribution of SLR and TC climatology change to future flood levels (Figure 4). This is indicated in the first paragraph in Discussion.

To summarize, we recommend our findings to be used as a guide to identify regions that future flood levels could surpass the typical height of flood protection measures and, thus, to prioritize

regions that need a regional/local-scale flood hazard study (which accounts for elements that are missing in basin-scale studies). We have added a following second paragraph on a comparison with the USACE study, which enhances the point as the USACE study is developed for engineering applications.

Minor Comments

Line 25: Should this not be: “Coastal flooding responds to the impacts of climate change, including sea level rise (SLR).”? SLR is an outcome of climate change and the two are not necessarily separated. However, one could make an argument that relative sea level rise could be separate when considering local subsidence.

Response: We agree that eustatic sea level rise is a consequence of climate change. We revised line 25. The sentence now reads as “Coastal flooding responds to both sea level rise (SLR) and storm climatology change.”

Line 58: It looks like you also do the U.S. Gulf Coast as well (Line 91, for example). Make that clear in the manuscript.

Response: The revised manuscript now clearly mentions that “Gulf Coast” is included. Also, we slightly revised the title of the manuscript to address the reviewer’s comment. Please see the revised manuscript.

Lines 354-357: A higher resolution mesh that covers the coastal floodplain will also likely result in changes to the results. I suggest briefly commenting on this.

Response: We agree that missing floodplains can have impacts on water levels. This is now mentioned in the first paragraph of “Limitations” in the revised manuscript.

Line 357-361: I suggest citing additional articles here that show the nonlinearities of SLR-surge are important, and not just those that show they are not substantial. A few examples are:

- Bilskie, M. V., et al. (2014). "Dynamics of sea level rise and coastal flooding on a changing landscape." *Geophysical Research Letters* 41(3): 927-934.
- Bilskie, M. V., et al. (2014). "Dynamics of sea level rise and coastal flooding on a changing landscape." *Geophysical Research Letters* 41(3): 927-934.
- Bilskie, M. V., et al. (2019). "Development of Return Period Stillwater Floodplains for the Northern Gulf of Mexico under the Coastal Dynamics of Sea Level Rise." *J. Waterway, Port, Coastal, Ocean Eng.* 145(2): 04018043.
- Atkinson, J. H., et al. (2013). "Sea-level rise effects on storm surge and nearshore waves on the Texas coast: influence of landscape and storm characteristics." *Journal of Waterway, Port, Coastal, and Ocean Engineering* 139(2): 98-117.

Response: Thanks for the suggestion. We found the suggested references informative. The references are cited in the revised manuscript.

Reviewers' comments:

Reviewer #1 (Remarks to the Author):

I thank the authors for their detailed replies to my comments. All of my comments have been addressed and I recommend to publish the article in its current form.

Best regards,
Andreas Prein

Reviewer #3 (Remarks to the Author):

Manuscript: Climate Change Exacerbates Hurricane Flood Hazards Along U.S. Atlantic Coast in Spatially Varying Patterns

Major Comments

Major Comment #1 from my original review (“I believe that studies, such as the one presented ...”) was not addressed in the manuscript itself. I appreciate the response; however, my comment was suggesting that you add material/discussion into the manuscript. I do appreciate the paragraph added regarding the USACE study and how it relates to your study. Please add portions of your response to this comment to the manuscript.

Minor Comments

Line 53: Change to “... a large number of simulations ...”

Line 53-54: Be more specific when you mean “costly numerical meshes.” I know what you mean in regard to computational cost, but I suggest you make this subtle change in the manuscript. Line 280: Change “significantly” to “substantially.” Significantly should be used sparingly in this context as it relates to statistical significant, which is not what are pointing out here.

Lines 285-287: These are very strong statements and I would caution around how this is framed. While it is correct that FEMA does not consider future effects of climate change, such as SLR, in their current framework, it is important to recognize the mission of the flood insurance studies. Therefore, I would add some context/discussion on why future changes are not currently part of their approved methods. FEMA cannot easily adopt new methods. New methods must go through a rigorous approval process and this can take some time.

In addition, the studies must be use equivalent methods for consistency. With that said, I do agree that future changes should be taken into account as new FEMA studies are being planned. In addition, you mention “some coastal communities are re-doing” FEMA flood maps while accounting for future changes, but only one citation is provided. It would be good to add additional citations that support this sentence. If not, I would soften the statement.

Line 423-425: Although computationally expensive, this approach was done in:

Bilskie, M. V., S. C. Hagen, and J. L. Irish (2019), Development of Return Period Stillwater Floodplains for the Northern Gulf of Mexico under the Coastal Dynamics of Sea Level Rise, *J. Waterway, Port, Coastal, Ocean Eng.*, 145(2), 04018043.

Of course the above study did not account for as large of a coastline as the current manuscript. However, the mesh used in Bilskie et al. contained 5.5 million nodes and shows that the capability does exist to simulate a large number of simulations with a suite of future climate change scenarios. Therefore, I would soften the claim made on lines 423-425 that it is impractical. Such studies do require careful consideration of the scenarios simulated and do require a very large computational cost.

Reviewer #1 (Remarks to the Author):

I thank the authors for their detailed replies to my comments. All of my comments have been addressed and I recommend to publish the article in its current form.

Best regards,
Andreas Prein

Reviewer #3 (Remarks to the Author):

See the attached PDF.

-Matt Bilskie

Manuscript: Climate Change Exacerbates Hurricane Flood Hazards Along U.S. Atlantic Coast in Spatially Varying Patterns

Major Comments

Major Comment #1 from my original review (“I believe that studies, such as the one presented ...”) was not addressed in the manuscript itself. I appreciate the response; however, my comment was suggesting that you add material/discussion into the manuscript. I do appreciate the paragraph added regarding the USACE study and how it relates to your study. Please add portions of your response to this comment to the manuscript.

Response: The reviewer was asking about the differences and relationships between the large-scale studies as in this manuscript and more detailed regional- and local-scale modeling. We explained it in the response but didn't add the details to the manuscript. Per the reviewer's suggestion, we have added portion of our previous response to a new paragraph (the current fourth paragraph in Discussion).

Minor Comments

Line 53: Change to “... a large number of simulations ...”

Response: Corrected.

Line 53-54: Be more specific when you mean “costly numerical meshes.” I know what you mean in regard to computational cost, but I suggest you make this subtle change in the manuscript.

Response: Changed to “computationally expensive numerical meshes”.

Line 280: Change “significantly” to “substantially.” Significantly should be used sparingly in this context as it relates to statistical significant, which is not what are pointing out here.

Response: Changed.

Lines 285-287: These are very strong statements and I would caution around how this is framed. While it is correct that FEMA does not consider future effects of climate change, such as SLR, in their current framework, it is important to recognize the mission of the flood insurance studies. Therefore, I would add some context/discussion on why future changes are not currently part of

their approved methods. FEMA cannot easily adopt new methods. New methods must go through a rigorous approval process and this can take some time. In addition, the studies must be use equivalent methods for consistency. With that said, I do agree that future changes should be taken into account as new FEMA studies are being planned.

In addition, you mention “some coastal communities are re-doing” FEMA flood maps while accounting for future changes, but only one citation is provided. It would be good to add additional citations that support this sentence. If not, I would soften the statement.

Response: We agree. In the current third paragraph of Discussion, we have changed the sentence to “Current flood risk mapping from the U.S. Federal Emergency Management Agency (FEMA) have not accounted for the effects of climate change.” to simply state the fact. Then we point out that our results indicate the flood risk will greatly change in the future, thus we suggest to account for the effect of SLR and storm change. Indeed, FEMA mapping is more complicated and detailed discussion is beyond the scope of this paper. To indicate that, we mention “Regional-scale (and local-scale) studies and flood mapping require a more detailed variation of flood levels along the coastlines to support regional flood mitigation strategies.” in the following paragraph when discussion about regional-scale modeling. We have also removed the point of “some coastal communities are re-doing” FEMA flood maps, as indeed we didn’t find additional references.

Line 423-425: Although computationally expensive, this approach was done in:

Bilskie, M. V., S. C. Hagen, and J. L. Irish (2019), Development of Return Period Stillwater Floodplains for the Northern Gulf of Mexico under the Coastal Dynamics of Sea Level Rise, J. Waterway, Port, Coastal, Ocean Eng., 145(2), 04018043.

Of course the above study did not account for as large of a coastline as the current manuscript. However, the mesh used in Bilskie et al. contained 5.5 million nodes and shows that the capability does exist to simulation a large number of simulations with a suite of future climate change scenarios. Therefore, I would soften the claim made on lines 423-425 that it is impractical. Such studies do require careful consideration of the scenarios simulated and do require a very large computational cost.

Response: We have changed “computationally impractical” to “computationally expensive, if possible, for full probabilistic assessments,...”, and we have added that “(Direct simulations may be applied for studies with substantially reduced number of scenarios, for example, through focusing on a few selected SLR scenarios)”, citing again the reference.

We thank the reviewer for going through our manuscript again and providing helpful suggestions.

REVIEWERS' COMMENTS:

Reviewer #3 (Remarks to the Author):

I thank the authors for their replies to my comments. All of my comments have been addressed and I recommend to publish the article in its latest form.